# Identification of glutathione transferase (GST P1) inhibitors via a high-throughput screening assay and implications as alternative treatment options for breast cancers

Sarah A. P. Pereira[1,2], Jonathan Vesin[3], Marc Chambon[3], Gerardo Turcatti[3], M. Lúcia M. F. S. Saraiva[1], Paul J. Dyson[2]*

1 LAQV, REQUIMTE, Departamento de Ciências Químicas, Faculdade de Farmácia, Universidade do Porto, Rua Jorge Viterbo Ferreira, Porto, Portugal, 2 Institut des Sciences et Ingénierie Chimiques, École Polytechnique Fédérale de Lausanne (EPFL), Lausanne, Switzerland, 3 Biomolecular Screening Facility, Ecole Polytechnique Federale de Lausanne (EPFL), Lausanne, Switzerland

* paul.dyson@epfl.ch

## Abstract

Glutathione S-transferase P1-1 (GST P1) is an important drug target as it is implicated in drug resistance. GST P1-1 inhibitors are typically non-productive analogues of glutathione which covers broad chemical space; hence it is likely that several clinically used drugs may unknowingly act as GST P1-1 inhibitors. We developed a high-throughput screening assay for GST P1-1 and screened 5830 compounds. From the screening, 24 potent GST P1-1 inhibitors were identified and assessed for cytotoxicity in MCF-7 and MDA-MB-231 breast cancer cell lines. Ethacrynic acid (a known GST P1-1 inhibitor), ZM 39923, PRT 4165, 10058-F4, and crypto-tanshinone were shown to be the most active. A competitive GST P1-1 assay was performed to identify the inhibition type of these five compounds. Another *in vitro* cytotoxicity experiment was conducted to explore the differences in the cytotoxicity profiles of the combinations tested. Western blot analysis was used to identify the presence of GST P1-1 in the breast cancer cell lines tested. The implications of these results concerning alternative treatment options for breast cancers are discussed.

## Introduction

Glutathione S-transferases (GSTs) are phase II detoxification enzymes, catalyzing glutathione (GSH) conjugation to xenobiotics, including drug compounds, to increase solubility and facilitate excretion, and have been identified as a drug target for many diseases [1]. The broad spectrum of substrates means GST activity has been implicated as a determinant of efficacy for many drugs [2,3], with isoenzyme, GST P1-1 most commonly associated with both innate and acquired drug resistance during cancer chemotherapy [1,4,5], including 5-fluorouracil- and cisplatin-resistant gastric

**Data availability statement:** All relevant data are within the paper and its Supporting Information files.

**Funding:** Swiss National Science Foundation and PT national funds (FCT/MCTES, Fundação para a Ciência e Tecnologia and Ministério da Ciência, Tecnologia e Ensino Superior) through grant UIDB/QUI/50006/2020 and UIDP/50006/2020. S. A. P. P. acknowledges FCT for her Ph.D. grant (SFRH/BD/138835/2018).

**Competing interests:** NO authors have competing interests.

cancers [6] and doxorubicin-resistant prostate cancer [7], because of overexpression in drug-resistant cell lines [8,9]. However, it is not clear if the observed resistance is related to the GSH conjugation activities or another function of GST P1-1. For example, GST P1-1 has been linked to adriamycin-resistant breast cancer, but adriamycin is a poor GST P1-1 substrate and glutathione conjugates of adriamycin do not occur under physiological conditions [10]. Alternatively, GST P1-1 may mediate its effect by sequestering the drug compounds, as recently identified for cisplatin [11]. Indeed, these enzymes were first identified for their ligand-binding properties [12]. GST P1-1 binding properties have also been linked to the regulation of intracellular processes, for example, binding to tumor necrosis factor receptor-associated factor 2 blocking apoptosis [13] and regulating mitogen-activated protein kinases through protein-protein interactions [10,14–16].

Despite the promise of GST P1-1 as a therapeutic target and the considerable research effort in developing GST inhibitors, to the best of our knowledge, few have entered clinical studies. One of the principal reasons for the lack of successful clinical applications is the ubiquitous nature and diverse function of GSTs, such that systemic inhibition could trigger unwanted side effects [17,18]. Indeed, ethacrynic acid (EA), one of the first and arguably the most widely studied inhibitors of GST P1-1 [19], is not used in the clinic for cancer therapy but is a clinically approved diuretic [20].

Based on the potential of GST P1-1 inhibitors in cancer chemotherapy [21], we examined other known drug compounds using high-throughput screening to identify putative drug compounds that target GST P1-1. Hit compounds identified from the high-throughput screen were validated via dose-response inhibition assays and further evaluated for cytotoxicity in MCF-7 estrogen and progesterone receptor-positive breast cancer cells and MDA-MB-231 triple-negative breast cancer cells. Lead compounds were characterized in competitive binding studies and the impact of the compounds on GST P1-1 expression was assessed. A schematic representation of the workflow is presented in Fig 1, and this study's implications with respect to drug repurposing are discussed. Note that high-throughput screening has been previously used to identify GST P1-1 inhibitors [22,23], but these former screens were limited to 800 compounds. The assay proposed in this manuscript allows large numbers of compounds to be screened, using small amounts of reagents, minimizing handling steps, increasing the overall efficiency of automation.

## Materials and methods

Recombinant human GST P1-1, glutathione reduced form, sodium fluoride, EDTA sodium salt, and glycerol were purchased from Sigma-Aldrich®. 1-Chloro-2,4-dinitrobenzene (CDNB) 98%, tris(hydroxymethyl)aminomethane hydrochloride, 99.5%, sodium dihydrogen phosphate dihydrate, and dithiothreitol 98% were purchased from abcr GmbH & Co. Di-sodium hydrogen phosphate heptahydrate and dimethyl sulfoxide were purchased from Merck® and absolute ethanol 99.8% was purchased from Thermo Fisher Scientific. Sodium chloride was purchased from Fluka®. The compounds used in the initial screen comprise the Prestwick Chemical Library® (Prestwick-Chemical, Illkirch Graffenstaden, France) 1280 Drug Collection,

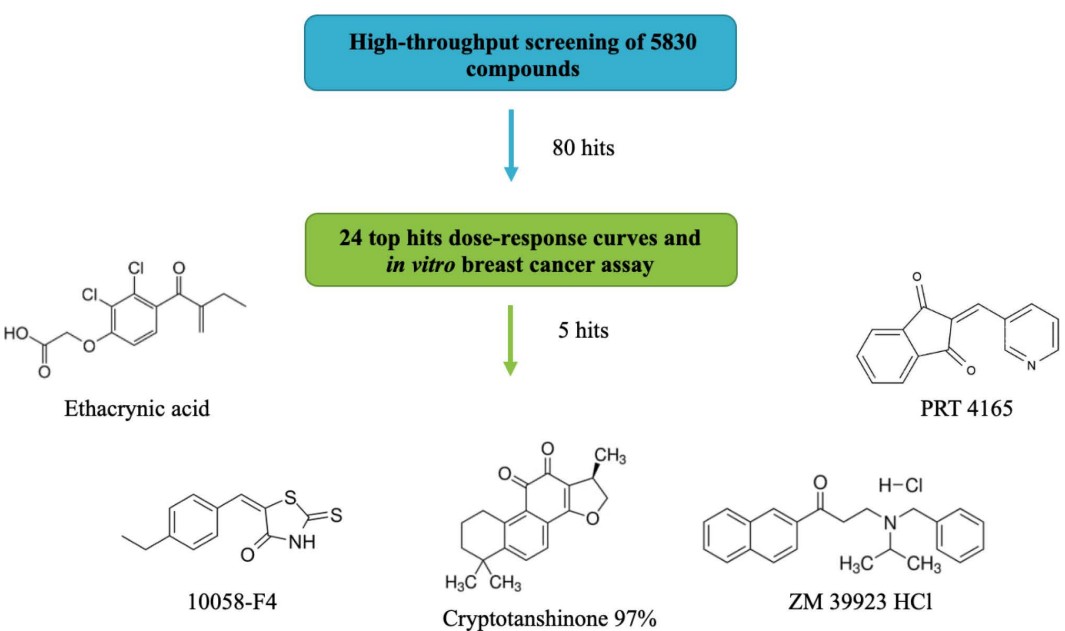

**Fig 1. Workflow used to identify and validate effective GST P1-1 inhibitors from a library of 5830 compounds, including all FDA-approved drugs.**

Repurposing collection (4479 compounds in preclinical, phases 1, 2 and 3 of clinical studies and approved from different suppliers mainly MedChem Express® and Tocris Bioscience®) and anticancer drugs plate (the 71 anticancer drug compounds used in the clinic were purchased from different suppliers including Acros Organics®, MedChem Express®, and Cayman Chemicals®). All drug compounds were supplied as 10 mM stock solutions in DMSO. Compounds were flushed with argon before being stored in the dark at −20°C using an automated storage system (A3-RTS store from Brooks) and their chemical integrity was controlled regularly by RP-HPLC, coupled to ESI-MS with UV and CAD detectors (Thermo).

Adherent mouse fibroblast connective tissue cells (L929) and human breast cancer cells (MCF-7 and MDA-MB-231) were obtained from ATCC, Middlesex, UK, and the European Collection of Authenticated Cell Cultures (ECACC), Public Health England, Salisbury, UK, respectively. DMEM GlutaMAX media (where DMEM = Dulbecco's modified Eagle medium) and Penicillin-streptomycin were obtained from Life Technologies, and heat-inactivated fetal bovine serum (FBS) was obtained from Sigma. White with clear bottom 96-well and 384-well microplates (cat. number 655180 and 781801, respectively) were obtained from Greiner Bio-One.

Pierce™ BCA protein assay kit was obtained from Thermo Scientific®, WesternBright™ ECL-spray (K-12049-D50) was obtained from Advansta®, GSTP1 monoclonal antibody, clone 3F2C2 and GADPH antibody goat anti-rabbit IgG H + L HRP were obtained from Merck®, Amersham™ Protran™ Premium Nitrocellulose blotting membrane 0.45 μm was obtained from Cytiva®.

## GST P1-1 assay optimization

The assay method was as described in Pereira *et al*, 2022, [24] with the following modifications. The GST P1-1 concentration used in the assays was 0.02 U mL⁻¹.

1 mM GSH final concentration was used. The GST P1-1 was pre-incubated with the GSH for 20 minutes at room temperature before the addition of 12.5 μL of 4 mM solution of CDNB. The reaction rate was recorded against time using UV/VIS spectrophotometer (SpectraMax M5e from Molecular devices) at 340 mm.

The reaction rate was controlled for the effect of solvents DMSO (0.08–5%) and ethanol (1–20%) and no significant effect was observed under the experimental conditions.

The parameters optimized for GST P1-1 activity were tested in a 96-well plate assay and the selected values are presented in Table 1. The GST P1-1 enzymatic reaction is represented in Fig 2.

### High-throughput GST P1-1 inhibition assay

The 5830 compounds (see above) were dispensed into 38 barcoded 384-well plates, using an Echo 655 acoustic liquid handler (Beckman Coulter®). Each drug compound was added to the respective well at a volume of 80 nL of 10 mM stock solution in DMSO, yielding a final compound concentration of 10 µM and a final DMSO concentration of 0.1%. Each plate's first and last two columns were used as negative (no compound added – 80 nL per well of DMSO) and positive (80 nL of EA added with a final concentration of 50 µM) controls, respectively. The Gyger Certus Flex dispenser was set up using a freshly prepared 1 mM of GSH and 20 nM of GST P1-1, pH 7.4, in one syringe and 8 mM CDNB solution in pure ethanol in the other syringe. Into each well 80 µL of GST-GSH solution was dispensed and incubated at 25°C for 20 minutes before the reaction was initiated by the addition of 4 µL of CDNB solution. The plates were incubated for 2 h in the dark at 25°C and then the absorbance was of each well recorded on a BioTek Synergy NEO HTS multimode plate reader (Agilent®) at a wavelength of 340 nm.

The z'factor was calculated for each plate according to Zhang et al. [25]. The Z-factor is a dimensionless, simple statistical characteristic for each HTS assay. This provides a valuable tool for the comparison and assessment of the quality of assays, and can be utilized in assay optimization and validation [25]. For each plate, the concentration of product was normalized by assigning a score of 0 (0%) to the negative control (average of maximal concentration of product measured in the presence of enzyme without any inhibitor) and a score of 1 (100%) was assigned to the positive control (average minimal concentration of product measured in the presence of enzyme with the reference inhibitor, i.e., EA). The

**Table 1. Parameters tested in the 96-well plate GST P1-1 inhibition assay.**

| Parameter | Values tested | Selected value |
|---|---|---|
| CDNB concentration (mM) | 0.25, 0.5, 1, 1.5, 2, 4, 6, 8 | 4 |
| GSH concentration (mM) | 0.1, 0.25, 0.5, 1, 2, 4, 6, 8 | 1 |
| GST P1-1 concentration (U/mL) | 0.002, 0.007, 0.01, 0.02, 0.03, 0.04, 0.05, 0.06 | 0.02 |
| Pre-incubation time (min) | 0, 15, 20, 30 | 20 |
| DMSO percentage (%) | 0.08, 0.15, 0.3, 0.6, 1.25, 2, 2.5, 5 | 2 |
| Ethanol percentage (%) | 1, 1.25, 2.5, 5, 10, 20 | 5 |
| Temperature (ºC) | 25 and 37 | 25 |

**Fig 2. GST P1-1 enzymatic reaction.**

experiment was performed in duplicate and the score for each drug compound was calculated as an average ± standard deviation. A hit is defined as a compound having a score greater than zero plus three times the standard deviation of the negative control in both replicates. An internal laboratory information management system (LIMS) was used for basic data processing, management, visualization, and statistical hit validation.

## Determination of IC$_{50}$ for GST P1-1 enzyme

The IC$_{50}$ of the GST P1-1 enzyme was determined for the 80 drug compounds with the highest score in the GST P1-1 screening assay. The assay conditions were used for the screening assay and each drug compound was tested at 8 different concentrations (µM) starting at 30 µM final assay concentration with a half-log dilution and 2 replicates per concentration.

## Cytotoxicity assays

L929, MCF-7, and MDA-MB-231 cell lines were cultured in DMEM GlutaMAX media containing 10% heat-inactivated FBS and 1% (v/v) penicillin-streptomycin at 37°C and CO$_2$ (5%). Cytotoxicity was determined using a 3-(4,5-dimethyl 2-thiazolyl)-2,5-diphenyl-2H-tetrazolium bromide (MTT) assay. 100 µL aliquots and approximately 50'000 cells/well were added to each well of flat-bottomed 96-well plates and incubated for 24 h in a humidified chamber at 37°C under saturated air with a 5% CO$_2$ atmosphere. Stock solutions of 10 mM drug compounds were prepared in DMSO and were sequentially diluted with the cell culture buffer to give a final DMSO concentration of 0.5%. The solutions were added to the 96-well plates in 100 µL aliquots and 100 µL of buffer was added as a control to give a final drug compound concentration range of 0 – 100 µM. The plates were incubated for a further 72 h under the same conditions as before. MTT (10 µL, 5 mg/mL in Dulbecco's phosphate-buffered saline) was added to each well and the plates were incubated for a further 3 h under the same conditions as before. The culture medium was aspirated from each well and the purple formazan crystals, formed by the mitochondrial dehydrogenase activity of vital cells, were dissolved in 100 µL DMSO. The absorbance of the resulting solutions was measured at 570 and 630 nm using a SpectraMax M5e multimode microplate reader and SoftMax Pro software, version 6.2.2. As the absorbance is directly proportional to the number of surviving cells, the percentage of surviving cells was calculated from the absorbance of wells relative to the control. The reported IC$_{50}$ values are based on the mean from four independent experiments, each comprising three replicates per concentration.

## Determination of inhibitor binding mechanism

The drug compounds' inhibition mechanism was investigated using the kinetic GST P1-1 activity assay. Concentrations of CDNB ranged from 0–40 mM with the drug solution at a constant concentration of 5 µM. Experiments were performed in duplicate.

The Michaelis-Menten constant, $K_m$, and half maximum velocity $V_{max}$ were calculated using GraphPad Prism 9 software. To identify the type of inhibition the Michaelis-Menten equation was applied (Eq. 1):

$$v_0 = \frac{V_{max}\,[S]}{[S] + Km}$$

(1)

where $v_0$ is the initial velocity of the reaction, $V_{max}$ is the maximum velocity achieved by the reaction at maximum concentrations of substrate, $[S]$ is the concentration of substrate, and $K_m$ is the substrate concentration at which the reaction velocity is half of the $V_{max}$ (Michaelis-Menten constant). The Michaelis-Menten equation predicts a hyperbolic relationship between initial velocity and substrate concentration, describing the kinetic behavior of enzymes [26]. Evaluating the differences in $K_m$ and $V_{max}$ compared with the reaction without an inhibitor allows the competition type of an inhibitor to be identified. Competitive inhibitors present an increase in the $K_m$ and with $V_{max}$ unaffected. Non-competitive inhibitors present a decrease in $K_m$ and $V_{max}$ [27].

## Cytotoxicity competitive inhibitory assay

The same protocol and detection method was used before (MTT) to perform them.

This study was initiated with the $IC_{50}$ concentrations of both compounds and then successive dilutions of them were made, giving a crossing of 6 concentrations of both compounds tested.

## Western blot immunoassay

MCF-7 and MDA-MB-231 cells (320'000 cells per 60 mm petri dish) were incubated for 24 h in a humidified chamber at 37°C under saturated air with a 5% $CO_2$ atmosphere. The cells were rinsed with ice-cold PBS twice and lysed in 100 μL of buffer containing 50 mM Tris, 250 mM NaCl, 5 mM EDTA, 50 mM NaF, 1 mM $Na_3VO_4$, and 1% NP-40 for 30 min on ice. After mechanical suspension, brief sonication, and centrifugation (14,000 g for 20 min), the supernatant was collected and assayed for protein content using the BCA method [28]. Aliquots containing 5 μg of protein were heated in a water bath at 100°C for 5 min and electrophoresed on 10% SDS-polyacrylamide gels. Proteins were transferred onto nitrocellulose membranes, which were then blocked by incubation with 5% milk (Rapilait) for 1 h at room temperature. The membranes were incubated with an anti-GST P1 antibody 1:1000 (Fisher Scientific AG) overnight at 4°C in the dark. The next day, the primary antibody is removed, and the membranes are rising with PBS Tween 0.1% three times. Then, the membranes were incubated with a secondary goat anti-mouse horseradish peroxidase-conjugated anti-immunoglobulin-G 1:5000 (Invitrogen). Protein bands were detected using an enhanced chemiluminescence western blot detection system (WesternBright™ ECL-spray from Advansta®). The membrane was then stripped by rinsing the blot twice with PBS Tween 0.1% and reprobed following the same method with an anti-GAPDH antibody 1:1000 (Merck®) incubated for 2 h in place of the anti-GST P1 antibody.

## Results and discussion

### Identification and validation of potent GST P1-1 inhibitors

Inhibition of the reaction between CDNB and GSH in the presence of GST P1-1 has been extensively used to evaluate compounds as potential GST P1-1 inhibitors [24,29,30]. Hence, this reaction was used with the parameter range evaluated using an univariant method to maximize the absorbance signal, and the optimized values for the 96-well plate assay are listed in Table 1 (see also S1 Fig in the S1 File for optimization of the solvents used). Km of GSTP1−1 enzymatic reaction was calculated to be 8.4 mM and to ensure high GST P1-1 activity in the inhibition assay a 4 mM solution of CDNB was used (S2 Fig in S1 File).

In the next step, the GST P1-1 inhibition assay was enlarged from a 96- to a 384-well plate assay (see Materials and Methods, and Fig S3 in S1 File) leading to an optimized volume of 80 μL. The 384-well plate assay was validated with a Z'prime between 0.5 and 0.8. The Prestwick chemical library, Repurposing collection, and a selection of 71 anticancer drugs, totaling 5830 compounds and including most FDA-approved small molecule drugs, were assayed, with 336 compounds identified as hits from the screening. From these 336 hits, the 80 compounds with the highest scores were evaluated in dose-response assays, and from these assays, 24 compounds with known anticancer activity and with hillslopes < 4 (except for nastorazepide, CD437, MK-5108, and hypericin) were selected to perform the reconfirmation assay. The four compounds with higher hillslopes were also selected for the reconfirmation assay since their mechanism of action is relevant to breast cancer treatment [31–34] and specifically for CD437 and hypericin the $IC_{50}$ values obtained for GST P1-1 are among the lowest (S2 Table in S1 File). Notably, EA, the positive control of this experiment, was also considered a hit since at 10 μM, the score obtained was 0.85 presenting a good hillslope and $IC_{50}$ value for GST P1-1. These compounds' hill slopes, IC50 values, and dose-response curves are shown in Fig S4 and Table S2 in S1 File.

### Cytotoxicity studies

The cytotoxicity of the 24 validated compounds was assessed on human breast cancer (MCF-7 and MDA-MB-231) cell lines and non-tumorigenic adherent mouse fibroblast connective tissue (L929) cells using the MTT assay with an

incubation period of 72 h (Table 2). Note that GST P1 is not expressed in MCF-7 cells [35], but is expressed in more invasive MDA-MB-231 cells [36], see Fig 3.

Nastorazepide is inactive ($IC_{50}$ > 100 μM) in all three cell lines (see Table 2). Other compounds are not endowed with cancer cell selectivity, either displaying similar cytotoxicity to all three cell lines or being more cytotoxic to the L929 cell line compared to one or both of the breast cancer cell lines. From the 24 validated compounds, five showed a reasonable degree of selectivity towards the breast cancer cell lines, i.e., EA, 10058-F4, cryptotanshinone, ZM 39923, and PRT 4165. EA and 10058-F4 are highly selective and show remarkably similar cytotoxicity profiles, whereas the other three compounds show more limited cancer cell selectivity. 10058-F4 is a novel inhibitor of c-Myc, a regulator of several genes such as the catalytic subunit of telomerase, cyclin D1, p53, and lactate dehydrogenase A, which are involved in cell growth, proliferation, immortality, cell cycle progression, apoptosis and metabolism [37,38]. Previous studies have demonstrated that c-Myc overexpression is linked to tumorigenesis across various cancer types, including breast, colon, lung, and ovarian cancers [37,39,40]. Several clinical studies demonstrated the potent anticancer effect of 10058-F4, either as a single agent or in combination with well-known chemotherapeutic agents such as cisplatin, doxorubicin, and 5-fluorouracil [41,42]. 10058-F4 promotes an increase in the expression of thioredoxin-interacting protein in triple-negative breast cancer drug-resistant cells, which enhances reactive oxygen species synthesis and reverses doxorubicin-induced chemotherapy resistance [43].

**Table 2. $IC_{50}$ values of the 24 hit compounds in MCF-7, MDA-MB-231, and L929 cells following incubation for 72 h.**

| Compounds | $IC_{50}$ (μM) (average ± standard deviation) | | | $IC_{50}$ (μM) GSTP1 enzymatic reaction |
|---|---|---|---|---|
| | MCF-7 | MDA-MB-231 | L929 | |
| Ethacrynic acid, EA | 13 ± 1 | 14 ± 1 | > 100 | 2.4 |
| 10058-F4 | 14 ± 2 | 15 ± 4 | > 100 | 2.8 |
| Cryptotanshinone 97.0% | 0.29 ± 0.04 | 2.9 ± 0.3 | 4.4 ± 0.3 | 8.9 |
| ZM 39923 (hydrochloride) | 4.3 ± 0.1 | 2.6 ± 0.1 | 8.2 ± 0.6 | 1.5 |
| PRT 4165 | 59 ± 7 | 27 ± 6 | > 100 | 1.3 |
| ZM 449829 | 6.8 ± 0.2 | 10.3 ± 0.2 | 8.1 ± 0.4 | 1.4 |
| CD437 | 0.51 ± 0.05 | 0.74 ± 0.07 | 0.24 ± 0.03 | 0.3 |
| AS252424 | 38 ± 2 | 91 ± 3 | 24 ± 2 | 0.6 |
| AZD9496 | 14 ± 1 | 47 ± 5 | 36 ± 4 | 2.1 |
| GW3965 (hydrochloride) | 25 ± 1 | 54 ± 1 | 24 ± 2 | 0.4 |
| nTZDpa | 39 ± 2 | 95 ± 4 | 36 ± 3 | 0.95 |
| RQ-00203078 | 17 ± 2 | 51 ± 6 | 42 ± 6 | 3.1 |
| Hypericin | 8 ± 1 | 26 ± 3 | 19 ± 1 | 0.1 |
| Embelin | 11 ± 1 | 12 ± 1 | 14 ± 1 | 0.6 |
| en460 | 10 ± 1 | 19 ± 1 | 18 ± 2 | 1.1 |
| Hexachlorophene | 14 ± 1 | 9 ± 1 | 11 ± 1 | 0.4 |
| Avasimibe | 12 ± 1 | 16 ± 1 | 13 ± 1 | 0.4 |
| MK-5108 | 3.4 ± 0.8 | > 100 | 2.2 ± 0.5 | 3.2 |
| TCS PIM-1 1 | 74 ± 4 | > 100 | 36 ± 3 | 2.2 |
| PSB 06126 | 45 ± 3 | > 100 | 39 ± 4 | 0.3 |
| PD-118057 | 41 ± 4 | > 100 | > 100 | 0.3 |
| Nastorazepide | > 100 | > 100 | > 100 | 3.3 |
| 5-Methyl-2-phenyl-1,2-dihydropyrazol-3-one, 98% | > 100 | > 100 | > 100 | 0.8 |
| TCID | > 100 | > 100 | > 100 | 0.9 |

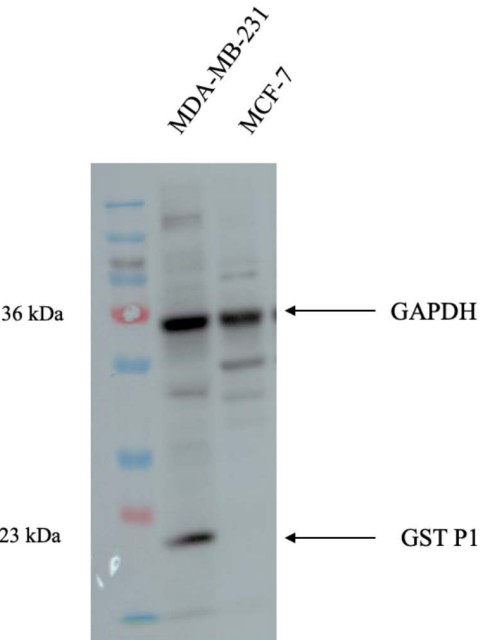

**Fig 3. Expression of GST P1 in MDA-MB-231 and MCF-7 cell lines.** GAPDH is a housekeeping protein used as an internal reference to normalize the expression of the target protein.

## Competitive inhibitory assay

The inhibition mechanism of the five most cytotoxic compounds, i.e., EA, 10058-F4, cryptotanshinone, ZM 39923, and PRT 4165, was determined using the previously optimized enzymatic 96-well plate method. EA and ZM 39923 were found to be competitive inhibitors whereas, 10058-F4, cryptotanshinone, and PRT 4165 are non-competitive inhibitors (Fig 4 and Table 3).

Combinations of inhibitors with competitive and non-competitive behavior were evaluated for cytotoxicity against MCF-7 and MDA-MB-231 cells to establish if synergistic activity could be achieved, i.e., if reduced doses of two combined compounds could achieve a similar effect to an individual compound. The $IC_{50}$ concentrations of both compounds were dosed followed by successive dilutions, aiming to identify the doses at which the two drugs yield the $IC_{50}$ concentration. In MCF-7 cells, the largest synergy was observed between EA and PRT 4165. The $IC_{50}$ values of EA and PRT 4165 when used alone are 13 µM and 59 µM, respectively, and when used together achieve an $IC_{50}$ value at concentrations of 2.9 µM for EA and 13.1 µM for PRT 4165. The three other inhibitors (10058-F4, cryptotanshinone, and ZM 39923) also synergized with EA in MCF-7 cells, but to a lesser extent. In the MDA-MB-231 cell line, the largest synergy is observed between EA and cryptotanshinone. When dosed as monodrugs, the $IC_{50}$ concentrations of EA is 14 µM and cryptotanshinone is 2.9 µM. In combination, the $IC_{50}$ value is reached at concentrations of EA and cryptotanshinone of 3.3 µM and 0.68 µM, respectively, similar to the MCF-7 cell line showing that combining competitive and non-competitive GST P1-1 inhibitors is optimal. 10058-F4 and PRT 4165 also weakly synergize with EA, and other combinations are not synergistic. Notably, the EA/PRT 4165, EA/cryptotanshinone, and EA/10058-F4 combinations synergize in both cancer cell lines.

## Conclusions

High-throughput screening of 5830 compounds was used to identify inhibitors of GST P1-1, an enzyme overexpressed in certain cancers that is responsible for drug resistance. The screening identified 24 compounds, of which EA, 10058-F4,

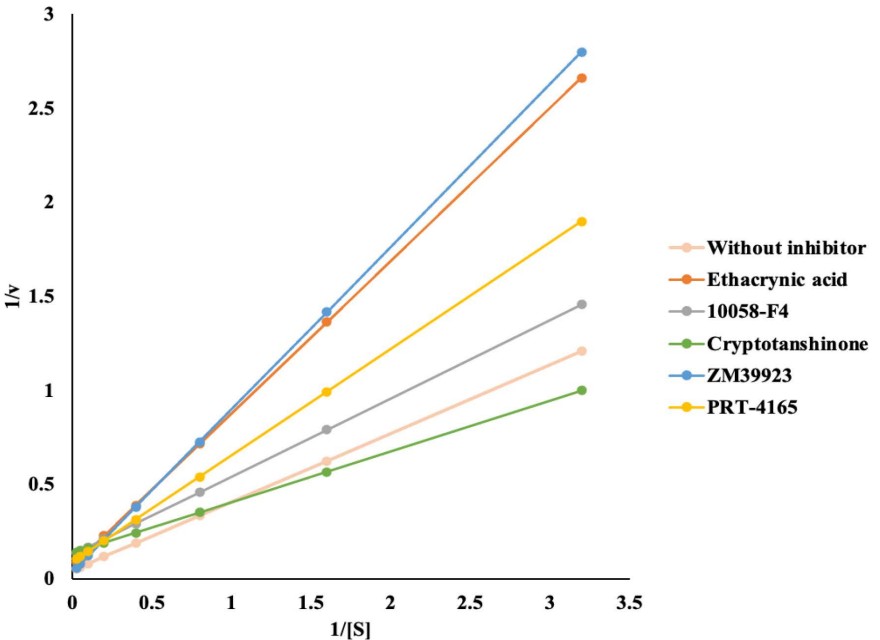

**Fig 4. Lineweaver-Burk plots (1/V versus 1/[S]) for the determination of kinetic parameters of GST P1-1 inhibition by the 5 most cytotoxic compounds obtained *in vitro*.**

**Table 3. Inhibition type according to the $K_m$ and $V_{max}$ obtained in the competitive GST P1-1 inhibition experiment.**

| Compound | Km | $V_{max}$ | Inhibition mechanism |
|---|---|---|---|
| Without inhibitor | 8.4 ± 2.4 | 23 ± 3 | – |
| Ethacrynic acid, EA | 12.3 ± 9.3 | 15.1 ± 7.3 | Competitive |
| 10058-F4 | 3.3 ± 1.8 | 7.9 ± 1.4 | Non-competitive |
| Cryptotanshinone | 2 ± 1 | 7.4 ± 1.3 | Non-competitive |
| ZM 39923 | 24.6 ± 6.7 | 28 ± 4 | Competitive |
| PRT 4165 | 6.4 ± 0.7 | 11.3 ± 0.5 | Non-competitive |

cryptotanshinone, ZM 39923, and PRT 4165 were cytotoxic to breast cancer cell lines while simultaneously displaying cancer cell selectivity. EA and ZM 39923 were identified as competitive GST P1-1 inhibitors, whereas PRT 4165, 10058-F4, and cryptotanshinone act as non-competitive GST P1 inhibitors. The inhibitors were screened in two-drug combinations and EA/PRT 4165, EA/cryptotanshinone, and EA/10058-F4 were found to exert synergistic cytotoxicity against both cancer cell lines. EA is the benchmark GST P1-1 inhibitor, whereas 10058-F4, cryptotanshinone, ZM 39923, and PRT 4165 are principally known to inhibit other targets. Notably, these potent GST P1-1 inhibitors display equivalent activity against the more invasive MDA-MB-231 cell line that expresses GST P1-1 compared with the MCF-7 cell line. In contrast, compounds that are poor inhibitors of GST P1-1 are often considerably less cytotoxic toward MDA-MB-231 cells (see Table 2). Hence, repurposing these potent inhibitors to cancers that overexpress GST P1-1, such as the triple-negative breast cancer studied here (MDA-MB-231), and potentially other cancers, could lead to promising clinical outcomes. It is not possible to identify unexpected side effects from our studies and future *in vivo* studies will be performed on promising inhibitors/combinations. Moreover, based on the extensive number of compounds screened in this study, the rational design of superior inhibitors should be feasible.

## Supporting information

**S1 File. Supplementary information.zip.**
(DOCX)

**S2 Fig. Original western blot plot.**
(TIF)

## Author contributions

**Data curation:** Sarah A.P. Pereira, Marc Chambon.

**Formal analysis:** Sarah A.P. Pereira.

**Funding acquisition:** Sarah A.P. Pereira, M. Lúcia M. F. S. Saraiva, Paul J. Dyson.

**Investigation:** Sarah A.P. Pereira, Jonathan Vesin.

**Methodology:** Sarah A.P. Pereira, Jonathan Vesin, Marc Chambon.

**Project administration:** Paul J. Dyson.

**Resources:** Jonathan Vesin, Marc Chambon, Gerardo Turcatti, Paul J. Dyson.

**Supervision:** M. Lúcia M. F. S. Saraiva, Paul J. Dyson.

**Validation:** Paul J. Dyson.

**Visualization:** Sarah A.P. Pereira.

**Writing – original draft:** Sarah A.P. Pereira.

**Writing – review & editing:** Jonathan Vesin, Marc Chambon, Gerardo Turcatti, M. Lúcia M. F. S. Saraiva, Paul J. Dyson.

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
