## [Decision Letter · Decision Letter 0]

Dear Dr. Pereira,

Thank you for submitting your manuscript to PLOS ONE. After careful consideration, we feel that it has merit but does not fully meet PLOS ONE’s publication criteria as it currently stands. Therefore, we invite you to submit a revised version of the manuscript that addresses the points raised during the review process.

**Please revise the article in the lights of comments of reviewers. **plosone@plos.org . A rebuttal letter that responds to each point raised by the academic editor and reviewer(s). You should upload this letter as a separate file labeled 'Response to Reviewers'.A marked-up copy of your manuscript that highlights changes made to the original version. You should upload this as a separate file labeled 'Revised Manuscript with Track Changes'.An unmarked version of your revised paper without tracked changes. You should upload this as a separate file labeled 'Manuscript'.

We look forward to receiving your revised manuscript.

Kind regards,

Abdul Rauf Shakoori

Academic Editor

PLOS ONE

Journal Requirements:

Inhibition of c-Myc using 10058-F4 induces anti-tumor effects in ovarian cancer cells via regulation of FOXO target genes - https://doi.org/10.1016/j.ejphar.2021.174345

(Among others)

In your revision ensure you cite all your sources (including your own works), and quote or rephrase any duplicated text outside the methods section. Further consideration is dependent on these concerns being addressed.

Swiss National Science Foundation and PT national funds (FCT/MCTES, Fundação para a Ciência e Tecnologia and Ministério da Ciência, Tecnologia e Ensino Superior) through grant UIDB/QUI/50006/2020 and UIDP/50006/2020. S. A. P. P. acknowledges FCT for her Ph.D. grant (SFRH/BD/138835/2018).

The authors that developed this work acknowledge the financial support from the Swiss National Science Foundation and PT national funds (FCT/MCTES, Fundação para a Ciência e Tecnologia and Ministério da Ciência, Tecnologia e Ensino Superior) through grant UIDB/QUI/50006/2020 and UIDP/50006/2020. S. A. P. P. acknowledges FCT for her Ph.D. grant (SFRH/BD/138835/2018).

Swiss National Science Foundation and PT national funds (FCT/MCTES, Fundação para a Ciência e Tecnologia and Ministério da Ciência, Tecnologia e Ensino Superior) through grant UIDB/QUI/50006/2020 and UIDP/50006/2020. S. A. P. P. acknowledges FCT for her Ph.D. grant (SFRH/BD/138835/2018).

7. PLOS ONE now requires that authors provide the original uncropped and unadjusted images underlying all blot or gel results reported in a submission’s figures or Supporting Information files. This policy and the journal’s other requirements for blot/gel reporting and figure preparation are described in detail at https://journals.plos.org/plosone/s/figures#loc-blot-and-gel-reporting-requirements and https://journals.plos.org/plosone/s/figures#loc-preparing-figures-from-image-files. When you submit your revised manuscript, please ensure that your figures adhere fully to these guidelines and provide the original underlying images for all blot or gel data reported in your submission. See the following link for instructions on providing the original image data: https://journals.plos.org/plosone/s/figures#loc-original-images-for-blots-and-gels.   

Reviewers' comments:

Reviewer's Responses to Questions

**Comments to the Author**

1. Is the manuscript technically sound, and do the data support the conclusions?

Reviewer #1: Yes

Reviewer #2: Yes

2. Has the statistical analysis been performed appropriately and rigorously?

Reviewer #1: Yes

Reviewer #2: Yes

3. Have the authors made all data underlying the findings in their manuscript fully available?

Reviewer #1: Yes

Reviewer #2: Yes

4. Is the manuscript presented in an intelligible fashion and written in standard English?

Reviewer #1: Yes

Reviewer #2: Yes

**Reviewer #1:**  GSTs (glutathione S-transferases) are known to contribute to cancer treatment resistance by helping cancer cells detoxify anticancer drugs. Inhibiting GSTP1-1 can potentially enhance the effectiveness of anticancer therapies.This approach leverages the inhibitory properties of ethacrynic acid against GSTP1-1 to potentially improve cancer treatment outcomes. By combining ethacrynic acid with other screened compound researchers aim to create molecules that specifically target breast cancer cells.

Please see the following comments and revise:-

High-throughput GST P1-1 inhibition assay?- How it is different from regular assay? what systems were used?

What is Z factor?

Western Blot assay should be changed to Western blot Immunoassay.

Figure 3 – Expression of GST P1 in MDA-MB-231 and MCF-7 cell lines. GAPDH is a housekeeping

protein used as an internal reference to normalize the expression of the target protein.

This figure should be modified and presented well with standard protein markers

Following paper should be cited in discusion with earlier work done.

a) Rasha Awni Guneidy, Eman Ragab Zaki, Nevein Salah-eldin Saleh, Abeer Shokeer, Inhibition of human glutathione transferase by catechin and gossypol: comparative structural analysis by kinetic properties, molecular docking and their efficacy on the viability of human MCF-7 cells, The Journal of Biochemistry, Volume 175, Issue 1, January 2024, Pages 69–83, https://doi.org/10.1093/jb/mvad070

Few sentences need check, repeatation of words etc are found or missing words (find the attachment)

**Reviewer #2:**  The manuscript entitled “Identification of glutathione transferase (GST P1) inhibitors via a high-throughput screening assay and implications as alternative treatment options for breast cancers” by Dyson et al. This study identifies 24 potent GST P1-1 inhibitors from a screen of 5,830 compounds, highlighting potential drug repurposing. Key inhibitors, including ethacrynic acid and cryptotanshinone, showed significant activity against MCF-7 and MDA-MB-231 cells. The manuscript is well-written; however there is a noticeable gap in the research. Before acceptance, the authors must address the following comments.

1. The manuscript is well-written; however, there are some overlapping words and punctuation errors in the text. The authors are advised to rectify these issues for enhanced clarity and coherence.

2. To enhance the manuscript's comprehensiveness, include discussions on potential limitations and future research directions.

3. Including relevant literature in the introduction will enrich the study's context and highlight its significance. Incorporating the following articles will provide valuable insights into cancer research, further supporting the study's focus.

https://doi.org/10.1038/s41598-017-10864-3

https://doi.org/10.1002/slct.202101853

https://doi.org/10.1016/j.ygyno.2013.02.005

**Do you want your identity to be public for this peer review?** For information about this choice, including consent withdrawal, please see our Privacy Policy

Reviewer #1: No

Reviewer #2: No

---

## [Author Response · Author response to Decision Letter 1]

31 Jan 2025

The response to the reviewers can be found in a file within the attach files section.

---

## [Editor Report · Decision Letter 1]

Identification of glutathione transferase (GST P1) inhibitors via a high-throughput screening assay and implications as alternative treatment options for breast cancers.

PONE-D-24-21813R1

Dear Dr. Pereira,

We’re pleased to inform you that your manuscript has been judged scientifically suitable for publication and will be formally accepted for publication once it meets all outstanding technical requirements.

Kind regards,

Abdul Rauf Shakoori, PhD

Academic Editor

PLOS ONE
---

## [Editor Report · Acceptance letter]

PONE-D-24-21813R1

PLOS ONE

Dear Dr. Pereira,

I'm pleased to inform you that your manuscript has been deemed suitable for publication in PLOS ONE. Congratulations! Your manuscript is now being handed over to our production team.

Kind regards,

on behalf of

Prof. Dr. Abdul Rauf Shakoori

Academic Editor

PLOS ONE